# Natural Element Static and Free Vibration Analysis of Functionally Graded Porous Composite Plates

## Jin-Rae Cho

Department of Naval Architecture and Ocean Engineering, Hongik University, Jochiwon, Sejong 30016, Republic of Korea; jrcho@hongik.ac.kr; Tel.: +82-44-860-2546

**Featured Application: Novel design of functionally graded ceramic–metal composite plates.**

**Abstract:** The static bending and free vibration of functionally graded (FG) porous plates were analyzed by a 2D natural element method (NEM). Recent studies on FG materials considered the porosity because micropores and porosity were observed during the fabrication of FG materials owing to the difference in solidification temperatures. However, the mechanical responses of FG porous plates were not sufficiently revealed, and furthermore most numerical studies relied on the finite element method. Motivated by this situation, this study intended to investigate the combined effects of material composition and porosity distributions and plate thickness on the static bending and free vibration responses of ceramic–metal FG plates using 2D NEM incorporated with the (3,3,2) hierarchical model. The proposed numerical method is verified from the comparison with the reference such that the maximum relative difference is 5.336%. Five different porosity distributions are considered and the central deflection and the fundamental frequency of ceramic–metal FG porous plates are parametrically investigated with respect to the combination of the porosity parameter, the ceramic volume fraction index, and the width–thickness ($w/t$) ratio and to the boundary condition. The ranges of three parameters were set to 0–0.5 for the porosity, 0–0.6 for the ceramic volume fraction, and 3–20 for the width–thickness ratio. It was found from the numerical experiments that the static and free vibration responses of ceramic–metal FG porous plates are significantly affected by these parameters.

**Keywords:** functionally graded; ceramic–metal porous composite plates; bending and free vibration; natural element method (NEM); porosity distribution; volume fraction distribution





## 1. Introduction

In the mid-1980s, the notion of functionally graded material (FGM) was proposed by Japanese scientists to develop high-performance heat-resisting composites for the Space Shuttle in high-temperature environments [1]. Thereafter, FGMs have been applied in various engineering fields, such as nuclear reactors and chemical plants, where high-performance heat-resisting materials are required. The heat-resisting composites used in the early days were lamination-type, which suffers from fatal thermal stress concentration at the lamina interfaces. This fatal stress concentration was caused by the sharp material discontinuity across the lamina interfaces and could frequently cause the microcracking and debonding. Owing to this problem, the conventional lamination-type composites encounter limitations in their use in high-temperature environments. The main idea of FGM was to enforce the material continuity across the lamina interfaces, which was achieved by introducing a graded layer between two homogeneous layers. The particles within the graded layer are mixed according to the desired volume fraction distribution with the gradient in a specific spatial direction, while satisfying the overall material continuity. Functionally graded volume fraction distribution within the graded layer is designed to maximize the target performance using the numerical optimization technique [2,3]. Due to

the unlimited potential for being a next-generation advanced material, FGMs have been spotlighted and have attracted extensive research. A tremendous number of papers on material characterization and modeling, analysis and design, and fabrication and testing have been published [4–6].

The most research on FGMs was performed based upon the implicit assumption that FGMs are perfect without porosity. However, it had been reported that micropores and porosity may be generated during the fabrication of FGMs owing to the difference in solidification temperatures of constituent materials [7–9]. Thereafter, a number of attempts have been made to model porous materials and investigate their mechanical responses. A simplified mixture rule was proposed to predict the material properties of porous materials and validated through the experiment of Ji et al. [10], where the predicted effective material properties were in good agreement with the experiment. Chen et al. [11,12] investigated static bending, buckling, free, and forced vibration responses of shear deformable FG porous beams using Timoshenko beam theory. Akbas [13] examined the effect of porosity on the static bending and free vibration of simply supported FG porous plates using the first-order shear deformation plate theory (SDPT). Ghadiri and SafarPour [14] investigated the free vibration of FG porous cylindrical micro shell subjected to thermal load using the first-order shear deformation shell and modified couple stress theories. Wu et al. [15] numerically investigated the free and forced vibration responses of FG porous beam-type structures by the finite element method (FEM) incorporated with Timoshenko and Euler–Bernoulli beam theories. She et al. [16] analyzed the nonlinear bending and vibration characteristics of FG porous tubes within the framework of nonlocal strain gradient theory. Gao et al. [17] proposed an analytical method for primary resonance analysis of FG porous cylindrical shells including the damping effect. Mirjavadi et al. [18] examined the effect of porosity on the nonlinear vibration of FG porous nano beams using von Kármán nonlinearity and the Euler–Bernoulli beam theory. Kim et al. [19] examined the bending, free vibration and buckling responses of FG porous micro plates using the first-order SDPT. Zhao et al. [20] investigated the free vibration of FG porous rectangular plates with uniform elastic boundary conditions using an improved Fourier series method. Ramteke et al. [21] examined the effect of grading pattern and porosity on the eigen characteristics of FG porous structure using higher-order displacement kinematics. Zenkour [22] presented a higher-order shear and normal deformation theory for the static analysis of porous thick rectangular plates by considering the thickness stretching effect. Kaddari et al. [23] proposed a new quasi-3D hyperbolic SDPT for the bending and free vibration analysis of FG porous plates on elastic foundation. Keleshteri and Jelovica [24] analyzed the large-amplitude free vibration of FG porous cylindrical panels considering different shell theories and boundary conditions. Gao et al. [25] parametrically examined the wave propagation behavior in FG metal porous plates reinforced with graphene platelets with respect to the distribution of porosity and graphene. Zghal et al. [26] numerically explored the influence of porosity on static bending analysis of FG porous beams using a refined mixed FE beam model.

Although many researchers studied the bending and free vibration responses of FG porous structures, the effects of porosity on these responses were not sufficiently investigated, particularly with regard to the combination of the material composition distribution, the porosity distribution, the width–thickness ratio, and boundary condition. Moreover, the numerical studies on FG porous structures mostly relied on the FEM. Therefore, deeper numerical investigation of these responses of FG porous structures using a reliable and effective meshfree method has significance. Motivated by this situation, the current study intended to deeply investigate the bending and free vibration responses of ceramic–metal FG porous plates using 2D NEM in the combination of abovementioned three parameters and to the boundary condition. As an extension of previous work [27,28] on the NEM hierarchical models for plates and sandwich plates, this study combines 2D NEM, the (3,3,2) hierarchical model and the cosine-type porosity distributions. As a last introduced meshfree method, NEM shows high numerical accuracy, even for coarse grids, thanks to the high smoothness of its Laplace interpolation (*L/I*) functions. Moreover, it can effec-

tively analyze 3D FG porous plates when it is incorporated with the hierarchical model, in which the displacement field is decomposed into the in-plane triple-vector function and 1D assumed thickness monomial.

In this study, the static bending and free vibration problem is formulated using a third-order SDPT-like (3,3,2) hierarchical model and approximated by 2D NEM. The thickness-wise distributions of porosity and constituent materials are expressed by the cosine and power-law functions, respectively. The numerical method is demonstrated through the numerical experiment and verified by comparison with the existing reference solutions. As well, the central deflections and the fundamental frequencies of ceramic–metal FG porous plates are parametrically investigated for the magnitude and distribution of porosity, the ceramic gradation index, the width–thickness ratio, and the boundary condition. The present numerical results are help one understand and design ceramic–metal FG porous plates by considering the porosity.

## 2. Functionally Graded Porous Plates

Figure 1a represents a full ceramic–metal FG porous plate with length $a$, width $b$ and thickness $h$, where the plate mid-surface is labeled by $\varpi \in \Re^2$. A Cartesian coordinate system $(O; x, y, z)$ sits at the upper left corner of $\varpi$ so that the plate top and bottom surfaces are positioned at $z = \pm h/2$. The composition of ceramic and metal particles varies through the thickness according to their volume fractions. Letting $V_c(z)$ and $V_m(z)$ be the volume fractions of ceramic and metal, both fractions satisfy the following constraint, given by

$$V_c(z) + V_m(z) = 1, \ -h/2 \le z \le h/2 \tag{1}$$

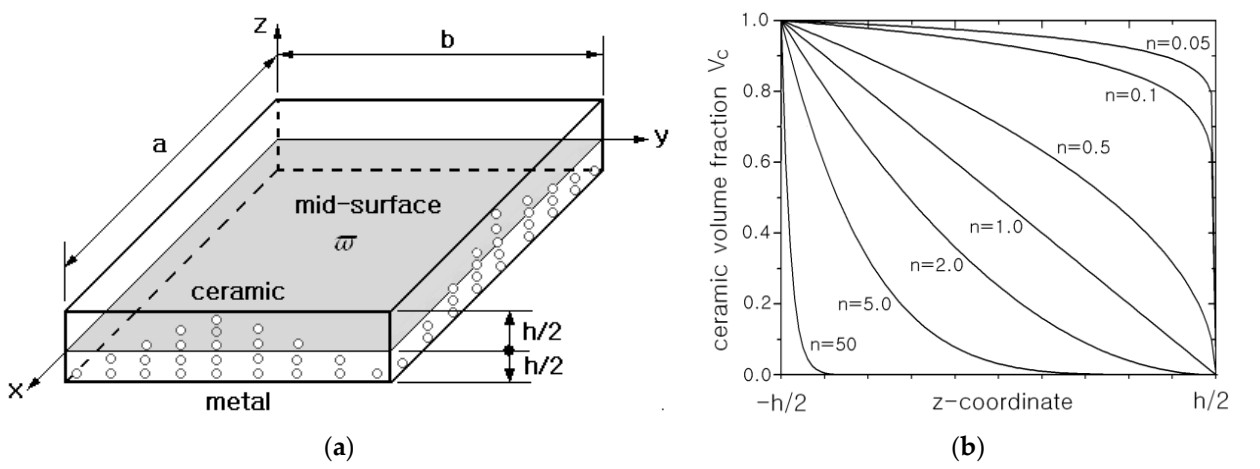

**Figure 1.** A ceramic–metal FG plate: (**a**) geometry and dimensions, (**b**) thickness-wise distributions of ceramic volume fraction $V_c$.

By virtue of this constraint, one can choose only one volume fraction to present two volume fractions within the plate. In the current study, the ceramic volume fraction $V_c(z)$ is selected and defined by

$$V_c(z) = (0.5 + z/h)^n, \ -h/2 \le z \le h/2 \tag{2}$$

with $n(n \ge 0)$ being the ceramic index. Figure 1b represents the thickness-wise ceramic volume fractions for different values of $n$, which were plotted using Excel. The plate becomes ceramic-rich as $n$ approaches zero and metal-rich as $n$ approaches infinity.

Figure 2 represents four different porosity distributions taken for the current study in which the porosity density changes in the $z$−direction only. Uniform and center-biased distributions are symmetric while lower- and upper-biased distributions are non-symmetric. In the current study, uniform distribution is called even, center-biased distribution is called

uneven or sym, and lower- and upper-biased distributions are called unsym-1 and unsym-2, respectively. First, the porosity $\psi(z)$ in even and uneven distributions is expressed by

$$\text{Even}: \ \psi(z) = e \tag{3}$$

$$\text{Uneven}: \ \psi(z) = e\left(1 - \frac{2|z|}{h}\right) \tag{4}$$

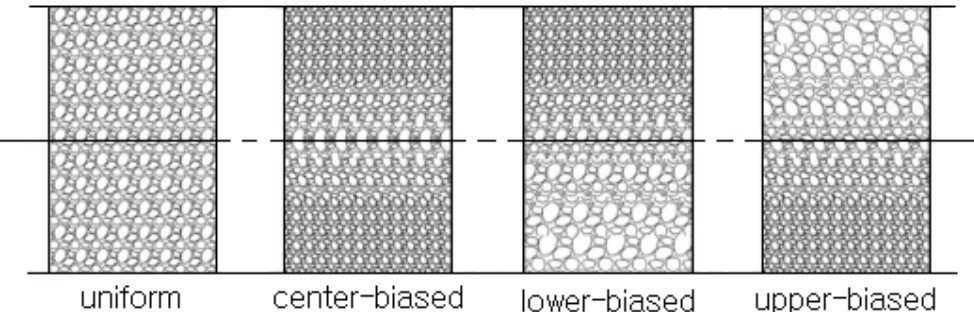

**Figure 2.** Uniform, center-, lower-, upper-biased porosity distributions [12].

Second, referring to a paper by Kim et al. [19], the porosity $\psi(z)$ in sym, unsym-1 and unsym-2 distributions is expressed by

$$\text{Sym}: \ \psi(z) = e\cos\left[\pi\left(\frac{z}{h}\right)\right] \tag{5}$$

$$\text{Unsym} - 1: \ \psi(z) = e\cos\left[\frac{\pi}{2}\left(\frac{z}{h} - 0.5\right)\right] \tag{6}$$

$$\text{Unsym} - 2: \ \psi(z) = e\cos\left[\frac{\pi}{2}\left(\frac{z}{h} + 0.5\right)\right] \tag{7}$$

using cosine functions. Here, $e(0 \leq e \leq 1)$ is the porosity parameter, and all the five porosity distributions have the same porous volume for a given value of $e$. According to the linear rule of mixtures, the equivalent material properties $\wp_{eff}(z)$ at any point within the FG porous plate is calculated by

$$\wp_{eff}(z) = [V_c(z)\wp_c + (1 - V_c(z))\wp_m](1 - \psi(z)) \tag{8}$$

in terms of two base material properties $\wp_c$ and $\wp_m$ of ceramic and metal, the volume fraction $V_c(z)$ and the porosity $\psi(z)$.

The mid-surface surface $\varpi \in \Re^2$ with its boundary $\partial\varpi$ can define a dimension-reduced 2D hierarchical model in which the displacement field $\boldsymbol{u}(x,y,z)$ is decomposed into $\boldsymbol{\Theta}(x,y) \cdot Q(z)$. Here, $\boldsymbol{\Theta}(x,y)$ and $Q(z)$ indicate the in-plane triple vector function and the assumed thickness monomial, respectively. The classical plate and shell theories and the hierarchical models [29–31] were developed according to this dimension reduction technique. Referring to reference [31], the (3,3,2) hierarchical model, which can be considered third-order SDPT, is expressed by

$$\left\{\begin{matrix} u_x \\ u_y \\ u_z \end{matrix}\right\}_{(x,y)} = \left\{\begin{matrix} u_x^0 \\ u_y^0 \\ u_z^0 \end{matrix}\right\}_{(x,y)} + \left\{\begin{matrix} u_x^1 \\ u_y^1 \\ u_z^1 \end{matrix}\right\}_{(x,y)} \times \left(\frac{2z}{h}\right) + \left\{\begin{matrix} u_x^2 \\ u_y^2 \\ u_z^2 \end{matrix}\right\}_{(x,y)} \times \left(\frac{2z}{h}\right)^2 + \left\{\begin{matrix} u_x^3 \\ u_y^3 \\ 0 \end{matrix}\right\}_{(x,y)} \times \left(\frac{2z}{h}\right)^3 \tag{9}$$

## 3. Natural Element (NE) Bending and Free Vibration Approximation

In the NE approximation, a uniform NEM grid is constructed on the mid-surface $\varpi$ of the FG porous plate. Referring to a previous paper [27], the NEM grid consists of a finite number of grid points called nodes and Delaunay triangles. For the NE approximation

of the (3,3,2) hierarchical model (9), its actual and virtual displacements $\boldsymbol{u}^h$ and $\boldsymbol{v}^h$ are expanded as ($\alpha, \beta = x, y, z$)

$$u_\alpha^h(\boldsymbol{x}) = \sum_{m=0}^{q_\alpha} \left( \sum_{J=1}^{N} U_{\alpha,J}^m \phi_J(x,y) \right) \cdot \left( \frac{2z}{h} \right)^m \tag{10}$$

$$v_\beta^h(\boldsymbol{x}) = \sum_{\ell=0}^{q_\beta} \left( \sum_{I=1}^{N} V_{\beta,I}^\ell \phi_I(x,y) \right) \cdot \left( \frac{2z}{h} \right)^\ell \tag{11}$$

with $q_x = q_y = 3$ and $q_z = 2$. Here, $\phi_J(x,y)$ are Laplace interpolation (*L/I*) functions [32,33] defined on the NEM grid and $U_{\alpha,J}^m$ indicates the nodal values to be determined.

Letting $q_x = q_y = q_z = q$ for the concise expression of mathematical formula, the virtual linear elastic strain vector $\boldsymbol{\varepsilon}\left( \boldsymbol{v}^h \right)$ and the actual Cauchy stress vector $\sigma\left( \boldsymbol{u}^h \right)$ are approximated as

$$\boldsymbol{\varepsilon}\left( \boldsymbol{v}^h \right) = \sum_{\ell=0}^{q} \sum_{I=1}^{N} \boldsymbol{L}_\varpi \phi_I V_I^\ell \cdot \left( \frac{2z}{h} \right)^\ell = \sum_{\ell=0}^{q} \sum_{I=1}^{N} \boldsymbol{B}_I^\ell V_I^\ell \cdot \left( \frac{2z}{h} \right)^\ell \tag{12}$$

$$\sigma\left( \boldsymbol{u}^h \right) = \boldsymbol{E}\boldsymbol{\varepsilon}\left( \boldsymbol{u}^h \right) = \boldsymbol{E} \sum_{m=0}^{q} \sum_{J=1}^{N} \boldsymbol{B}_J^m U_J^m \cdot \left( \frac{2z}{h} \right)^m \tag{13}$$

in which the gradient-like matrix $\boldsymbol{L}_\varpi$ and the partial differential matrix $\boldsymbol{B}_J^m$ are respectively defined by

$$\boldsymbol{L}_\varpi = \begin{bmatrix} \partial_{,x} & 0 & 0 & \partial_{,y} & 0 & m/z \\ 0 & \partial_{,y} & 0 & \partial_{,x} & m/z & 0 \\ 0 & 0 & m/z & 0 & \partial_{,y} & \partial_{,x} \end{bmatrix}^T \tag{14}$$

$$\begin{aligned} \boldsymbol{B}_J^m &= \boldsymbol{L}_\varpi \phi_J \\ &= \begin{bmatrix} \phi_{J,x} & 0 & 0 & \phi_{J,y} & 0 & m\phi_J/z \\ 0 & \phi_{J,y} & 0 & \phi_{J,x} & m\phi_J/z & 0 \\ 0 & 0 & m\phi_J/z & 0 & \phi_{J,y} & \phi_{J,x} \end{bmatrix}^T \end{aligned} \tag{15}$$

with $\partial_{,x} = \partial/\partial x$ and $\phi_{J,x} = \partial\phi_J/\partial x$. $\boldsymbol{E}$ denotes the ($6 \times 6$) matrix of the elastic modulus $E$ and the Poisson's ratio $\nu$.

Introducing Equations (12) and (13) into the weak form of static equilibrium [27] of FG porous plates leads to

$$[\boldsymbol{K}]_{IJ}^{\ell m} \{\boldsymbol{U}\}_J^\ell = \{\boldsymbol{F}\}_I^m \tag{16}$$

with the stiffness matrix $[\boldsymbol{K}]$ and the load vector $\{\boldsymbol{F}\}$ defined by

$$[\boldsymbol{K}]_{IJ}^{\ell m} = \int_{-h/2}^{h/2} \left[ \int_\varpi \left\{ \left( \boldsymbol{B}_I^T \boldsymbol{E}_1 \boldsymbol{B}_J \right)_{FI} + \left( \boldsymbol{B}_I^T \boldsymbol{E}_2 \boldsymbol{B}_J \right)_{RI} \right\} d\varpi \right] \cdot \left( \frac{2z}{h} \right)^{\ell+m} dz \tag{17}$$

$$\{\boldsymbol{F}\}_I^m = \int_{\partial\Omega_N} \tilde{t}_\alpha \phi_I ds \cdot (2z^*/h)^m dA \tag{18}$$

Here, $\tilde{t}$ indicates the external force acting on the natural boundary $\partial\Omega_N$ located at the vertical position $z^*$. The subscripts *FI* and *RI* in Equation (17) denote the full numerical integration using 7 Gauss points and the reduced integration using only 1 Gauss point. The reduced integration is used to overcome shear locking [34,35] for the bending-prevailed thin elastic structures, for which the material constant matrix $\boldsymbol{E}$ is divided into

$$[\boldsymbol{E}] = \begin{bmatrix} \boldsymbol{E}_1 & 0 \\ 0 & \boldsymbol{E}_2 \end{bmatrix}, \ [\boldsymbol{E}_1] = \begin{bmatrix} C_1 & C_2 & C_2 \\ C_2 & C_1 & C_2 \\ C_2 & C_2 & C_1 \end{bmatrix} \tag{19}$$

with $[E_2] = diag[G, G', G']$, in which $C_1 = (1 - v)E/[(1 - 2v)(1 + v)]$, $C_2 = vE_1$, $G = E/2(1 + v)$ and $G' = G/\kappa$ with the shear correction factor $\kappa = 6/5$.

For the free vibration $u(x; t) = \bar{u}(x) \cdot e^{j\omega t}$ of FG porous plate, the mass matrix $[M]$ is defined by

$$[M]^{\ell m}_{\alpha\beta, IJ} = \int_{-h/2}^{h/2} \left[ \int_\omega \rho(z)(\phi_I I)(\phi_J I) \, d\omega \right] \cdot \left( \frac{2z}{h} \right)^{\ell + m} dz \tag{20}$$

with the $(3 \times 3)$ identity matrix $[I]$ and the density $\rho(z)$. Then, the weak form of dynamic equilibrium of the FG porous plate leads to

$$\left[ K - \omega^2 M \right]^{\ell m}_{IJ} \left\{ \bar{u} \right\}^\ell_J = 0 \tag{21}$$

to compute the natural frequencies $\{\omega\}^\ell_J$ and the natural modes $\left\{ \bar{u} \right\}^\ell_J$.

## 4. Numerical Results

Figure 3a presents a metal–ceramic FG square plate under the action of uniform vertical distributed load $q_0 = 1.0 \, \text{N/m}^2$, where the side length $a$ is 0.1 m and the thickness $h$ are variables for the parametric investigation. The elastic modulus $E$, Poisson's ratio $v$ and the density $\rho$ are 70 GPa, 0.3 and 2707 kg/m$^3$ for metal and 380 GPa, 0.3 and 3800 kg/m$^3$ for ceramic. The simply supported condition is enforced as follows: $\overline{U}^0_z = 0$ for all the four sides, $\overline{U}^0_x = \cdots = \overline{U}^3_x = 0$ for two sides ① and ③, and $\overline{U}^0_y = \cdots = \overline{U}^3_y = 0$ for the other sides ② and ④. The full in-plane integration on the mid-surface for the stiffness and mass matrices and the load vector was performed using 7 Gauss points, while the thickness-wise integration was made using the trapezoidal rule by dividing the thickness into 50 uniform segments. The solutions of linear matrix Equation (16) were obtained by the frontal solver. The evaluation points and the calibrated deflection and stresses used in this study are defined by

$$\overline{w} = \frac{10E_c h^3}{q_0 a^4} w(0,0,0), \quad \overline{\sigma}_x = \frac{h}{q_0 a} \sigma_x(0,0,z^*), \quad \overline{\tau}_{xz} = \frac{h}{q_0 a} \tau_{xz}\left( -\frac{a}{2}, 0, z^* \right) \tag{22}$$

where, $z^*$ is the vertical position where the stresses show their maximum absolute values.

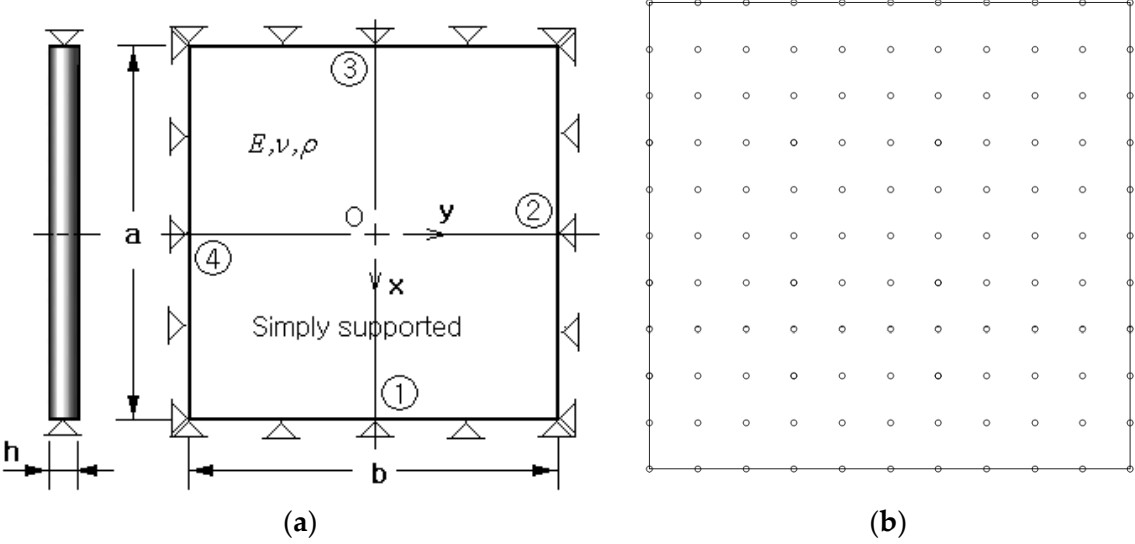

(a)  (b)

**Figure 3.** (a) A square FG porous plate under the uniform vertical distributed load $q_0$, (b) an $11 \times 11$ uniform NEM grid.

First, the variation in $\overline{w}$ and $\overline{\sigma}_x$ was examined with respect to the NEM grid density, which is illustrated in Figure 3b. The width–thickness ratio $a/h$ was 10 and the ceramic index $n$ and the porosity parameter $e$ were set by zero. The numerical results are presented in Table 1, where the values within the parentheses are the relative differences to the values obtained using the $19 \times 19$ NEM grid. It is seen that two calibrated quantities converge to the reference values proportional to the grid density and a $15 \times 15$ uniform grid shows the relative differences less than 1.0%, so according to this convergence test, a $15 \times 15$ uniform grid was chosen for the remaining numerical experiments, unless stated otherwise.

**Table 1.** Variation in $\overline{w}$ and $\overline{\sigma}_x$ with respect to the grid density ($a/h = 10$, (3,3,2) hierarchic model).

| Items | Grid Density | | | | |
|---|---|---|---|---|---|
| | 5×5 | 7×7 | 11×11 | 15×15 | 19×19 |
| $\overline{w}$ | 0.41165 (−14.631%) | 0.45044 (−6.586%) | 0.47284 (−1.941%) | 0.47943 (−0.574%) | 0.48220 (Ref.) |
| $\overline{\sigma}_x (\times 10^{-1})$ | 1.7186 (−54.557%) | 2.9076 (−13.118%) | 3.6306 (−4.001%) | 3.7449 (−0.978%) | 3.7819 (Ref.) |

The calibrated central deflection $\overline{w}$ was computed for various combinations of $a/h$, $n$ and $e$ and compared with those of Demirhan and Taskin [36] as given in Table 2. The reference solutions were analytically solved using a four-variable refined plate theory. It is seen that the present results coincide well with the reference solutions such that the peak relative difference is 5.336% at the combination of $a/h = 20, n = 0.5$ and $e = 0.2$ for the uneven porosity distribution.

**Table 2.** The calibrated central deflections $\overline{w}$ of simply supported square FG porous plate.

| a/h | n | e | Even | | Uneven | |
|---|---|---|---|---|---|---|
| | | | Present | Ref. [36] | Present | Ref. [36] |
| 10 | 0.5 | 0.0 | 0.71929 | 0.71361 | 0.71929 | 0.71361 |
| | | 0.2 | 0.89159 | 0.88950 | 0.77523 | 0.81751 |
| | | 0.4 | 1.18520 | 1.18947 | 0.97001 | 0.95489 |
| | 1.0 | 0.0 | 0.93275 | 0.92873 | 0.93275 | 0.92873 |
| | | 0.2 | 1.28790 | 1.29241 | 1.14190 | 1.13392 |
| | | 0.4 | 2.23613 | 2.29216 | 1.55447 | 1.58372 |
| 20 | 0.5 | 0.0 | 0.68489 | 0.69209 | 0.68489 | 0.69209 |
| | | 0.2 | 0.84938 | 0.86177 | 0.74850 | 0.79069 |
| | | 0.4 | 1.13005 | 1.15510 | 0.92341 | 0.92626 |
| | 1.0 | 0.0 | 0.88691 | 0.89968 | 0.88691 | 0.89968 |
| | | 0.2 | 1.22532 | 1.25611 | 1.09321 | 1.09915 |
| | | 0.4 | 2.12631 | 2.24360 | 1.45637 | 1.44418 |

In addition, the free vibration of the simply supported FG porous plate was performed using Lanczos transformation and Jacobi iteration methods. The lowest 10 natural frequencies for the previous combinations of $a/h$, $n$ and $e$ were computed using the $15 \times 15$ uniform NEM grid. In Table 3, the calibrated fundamental frequencies $\overline{\omega}_{(1,1)} = \omega_{(1,1)} h \sqrt{\rho_m / E_m}$ are compared with those in reference [36]. It is seen that the calibrated fundamental frequencies predicted by the present method coincide well with those of reference [36]. The detailed numerical values informs that the maximum relative difference is 3.954% at the combination of $a/h = 20, n = 1.0$ and $e = 0.4$ for the even porosity distribution. Hence, it has been justified that the present (3,3,2) hierarchical model is reliable for the free vibration analysis of FG porous plates. Thus, the present numerical method using the (3,3,2) hierarchical model has been verified such that thepeak relative difference between the present method and the reference is 5.336%.

**Table 3.** The calibrated fundamental frequencies $\overline{\omega}_{(1,1)}$ of simply supported square FG porous plate.

| $a/h$ | $n$ | $e$ | Even | | Uneven | |
|---|---|---|---|---|---|---|
| | | | Present | Ref. [36] | Present | Ref. [36] |
| 10 | 0.5 | 0.0 | 0.09702 | 0.09763 | 0.09702 | 0.09763 |
| | | 0.2 | 0.09682 | 0.09719 | 0.09896 | 0.10013 |
| | | 0.4 | 0.09598 | 0.09592 | 0.10120 | 0.10216 |
| | 1.0 | 0.0 | 0.08767 | 0.08796 | 0.08767 | 0.08796 |
| | | 0.2 | 0.08350 | 0.08324 | 0.08863 | 0.08859 |
| | | 0.4 | 0.07331 | 0.07194 | 0.08893 | 0.08991 |
| 20 | 0.5 | 0.0 | 0.02503 | 0.02475 | 0.02503 | 0.02476 |
| | | 0.2 | 0.02497 | 0.02463 | 0.02555 | 0.02542 |
| | | 0.4 | 0.02475 | 0.02429 | 0.02615 | 0.02606 |
| | 1.0 | 0.0 | 0.02262 | 0.02230 | 0.02262 | 0.02230 |
| | | 0.2 | 0.02154 | 0.02110 | 0.02282 | 0.02271 |
| | | 0.4 | 0.01893 | 0.01821 | 0.02299 | 0.02288 |

Next, the static bending and the free vibration of FG porous plates were parametrically examined for the major parameters. Figure 4a shows the variation of calibrated central deflection to the porosity parameter $e$ for the even and uneven porosity distributions when the $w/t$ ratio $a/h$ is 20. It is seen that the central deflection of even porosity distribution is larger than that of uneven porosity distribution, because not only the total amount of porosity is larger at the even porosity distribution but also the mid-surface-biased uneven porosity distribution provides the higher plate bending stiffness than the even porosity distribution. Meanwhile, the calibrated central deflection increases in proportion either to the ceramic index $n$ or to the porosity parameter $e$. Thisis because the plate bending stiffness becomes lower as either of the two indices becomes larger. This increase trend in $\overline{w}$ with increasing the porosity parameter $e$ is less apparent at the uneven porosity distribution, because the porosity is biased towards the mid-surface while the ceramic having the higher elastic modulus becomes more concentrated near the top surface proportional to the ceramic index. Figure 4b shows the variation in $\overline{w}$ to the porosity parameter for different $w/t$ ratios $a/h$, where the $\overline{w}$ uniformly decreases in proportion to $a/h$ because the central deflection is calibrated with the thickness, as given in Equation (22). Meanwhile, the relative difference in $\overline{w}$ between two porosity distributions is not shown to be affected by the value of $a/h$ because the thickness-wise distributions of metal and ceramic are not changed, even when the value of $a/h$ is changed.

Figure 5a represents the variation of $\overline{w}$ to the porosity parameter $e$ for different boundary conditions when the values of $a/h$ and $n$ are 20 and 0.3. C, S and F stand for clamped, simply supported and free, and the combined four charac-ters indicate a set of boundary conditions specific to the sides ①, ②, ③ and ④ of the FG porous plate shown in Figure 3a. It is observed that the lowest central deflection occurs at CCCC and the highest one is seen at SFSF. The second and third in the magni-tude of $\overline{w}$ are SSSS and SCSC, because the stronger the boundary condition is, the higher the plate bending stiffness is. Meanwhile, the relative difference in $\overline{w}$ between two porosity distributions is not seen to be affected by the boundary condition type. Figure 5b compares the variations of $\overline{w}$ between three cosine-type porosity distributions, where the sym shows the lowest level and the unsym 2 shows a lower level than the unsym 1, respectively. The FG plate with the sym porosity distribution is stiffer than the FG plates with unsym 1 and 2 porosity distributions because the porosity in the sym distribution is concentrated at the mid-surface. Meanwhile, the bending stiffness of FG plates having the unsym 2 porosity distribution is higher than that of FG plates with the unsym 1 porosity distribution. This is because the porosity in the former distribution is biased towards the top surface, but the ceramic with the higher elastic modulus is concentrated at the bottom surface when the ceramic index $n$ is 0.3.

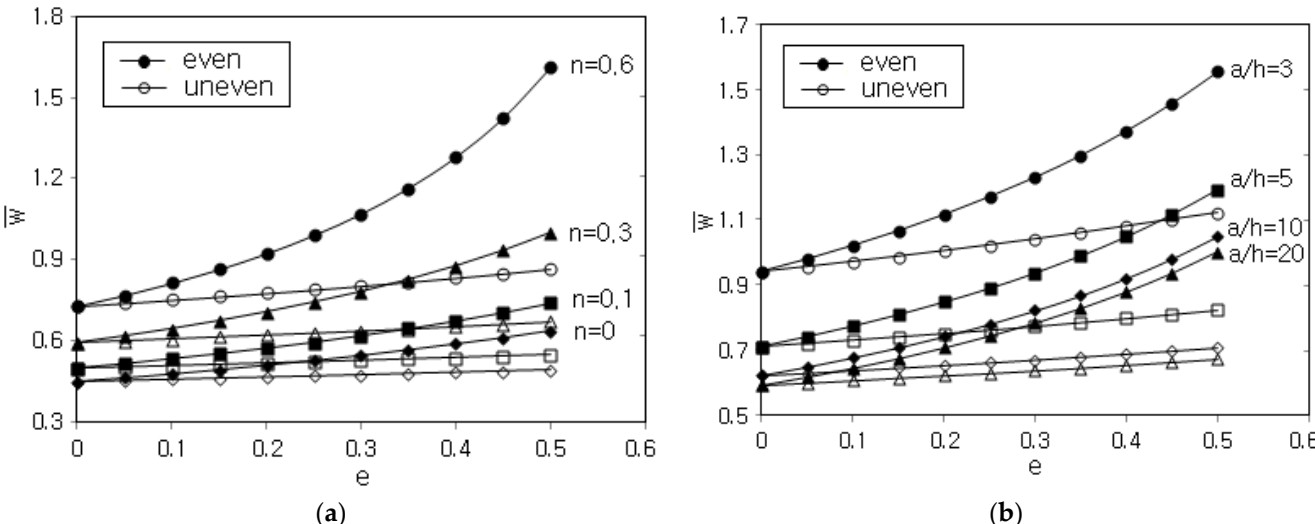

**Figure 4.** Dependence of the calibrated central deflection on the porosity parameter: (**a**) for different ceramic indices ($a/h = 20$), (**b**) for different width–thickness ratios ($n = 0.3$).

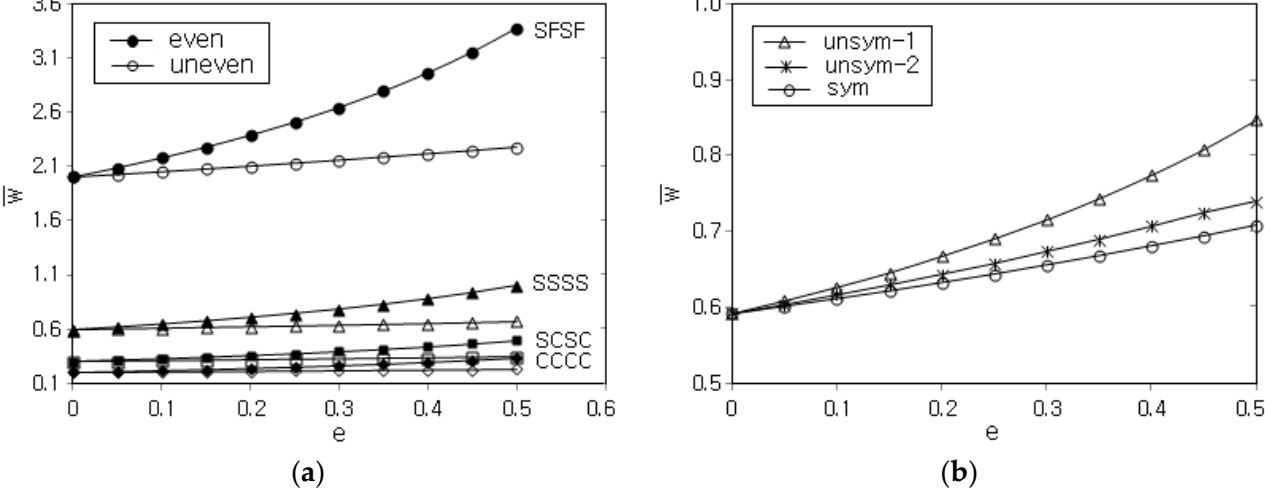

**Figure 5.** Dependence of the calibrated central deflection on the porosity parameter ($a/h = 20, n = 0.3$) (**a**) for different boundary conditions, (**b**) for different cosine-type porosity distributions.

Figure 6a compares the variation in $\overline{\sigma}_x$ to the porosity parameter $e$ for different porosity distributions. The peak value of $\overline{\sigma}_x$ occurs at the top of plate for all the three distributions and the unsym 1 distribution shows the highest level. The $\overline{\sigma}_x$ of unsym 2 distribution is lower than one of sym distribution, and furthermore it decreases in proportion to the porosity parameter $e$. This is because the elastic modulus at the top for the unsym 2 distribution is lower than one for the sym, and its decrease with increasing the value of $e$ is larger than the increase of $\overline{w}$ proportional to $e$. Figure 6b compares the variations of $\overline{\tau}_{xz}$ to the value of $e$ for three different porosity distributions. The peak of $\overline{\tau}_{xz}$ appears near the neutral surface regardless of the porosity distribution, and the highest level occurs at the unsym 1 distribution. The calibrated shear stresses of unsym 2 and sym distributions decrease proportionally to $e$ because the decrease inshear modulus with increasing $e$ is larger than the increase in $\overline{w}$ with the value of $e$.

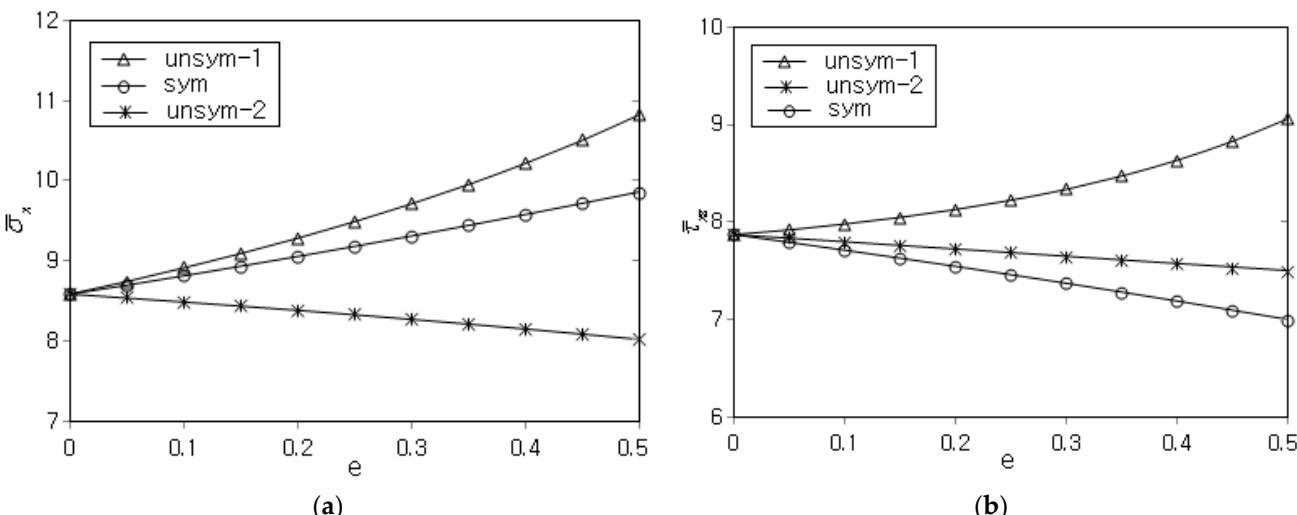

**Figure 6.** Dependence of the calibrated stress on the porosity parameter ($a/h = 20, n = 0.3$): (**a**) axial stress $\overline{\sigma}_x$, (**b**) shear stress $\overline{\tau}_{xz}$.

Figure 7a compares the variations of calibrated fundamental frequency $\overline{\omega}_{(1,1)}$ to the porosity parameter $e$ between the even and uneven porosity distributions, where the fundamental frequency is calibrated by $\overline{\omega}_{(1,1)} = \omega_{(1,1)} a^2 \sqrt{\rho_m/E_m}/h$. Similarly to the calibrated central deflection shown in Figure 4a, the $\overline{\omega}_{(1,1)}$ increases in proportion to $e$, except for the even distribution with $n = 0.6$. This is because the decrease in density with increasing $e$ gives a higher effect on the free vibration than the decrease inelastic modulus proportional to $e$ and vice versa for the exceptional case. Meanwhile, the uneven distribution provides the higher values of $\overline{\omega}_{(1,1)}$ than the even distribution when the ceramic index $n$ is $\geq 0.3$ since the porosity in the uneven distribution is biased near the mid-surface, while the ceramic with the higher elastic modulus becomes more concentrated at the bottom proportional to $n$. Figure 7b represents the variation in $\overline{\omega}_{(1,1)}$ to the porosity for different $w/t$ ratios $a/h$, where the $\overline{\omega}_{(1,1)}$ uniformly increases in proportion to $a/h$ owing to the calibration. According to this calibration, it is seen that the relative difference in $\overline{\omega}_{(1,1)}$ between even and uneven porosity distributions uniformly increases in proportion to $a/h$. The uneven porosity distribution provides higher fundamental frequencies than the even porosity distribution when the $a/h$ equals to and larger than 5.0. The relative difference in $\overline{\omega}_{(1,1)}$ between even and uneven porosity distributions uniformly increases with the ceramic index and the $w/t$ ratio. This is because the difference in thickness-wise material composition distributions between even and uneven porosity distributions becomes more apparent in proportion to these two factors.

Figure 8a compares the variations in $\overline{\omega}_{(1,1)}$ to the porosity parameter for different boundary conditions, where the highest frequency appears at CCCC and the second and third and lowest ones are shown at SCSC, SSSS and SFSF, respectively. This order of the magnitude of $\overline{\omega}_{(1,1)}$ is completely contrary to that of $\overline{w}$ shown in Figure 5a, which is consistent with the fact that the stronger the constraint is, the higher the fundamental frequency is. Figure 8b shows the variation in $\overline{\omega}_{(1,1)}$ to the porosity parameter for three different porosity distributions, where the order of the magnitude of $\overline{\omega}_{(1,1)}$ is completely contrary to that of $\overline{w}$ shown in Figure 5b. This is because the natural frequency and the central deflection show opposite variations to each other in their magnitudes to the structure stiffness. The unsym 1 distribution shows the lowest level and a decreasing trend for the porosity parameter, because both the porosity and the ceramic with the higher elastic modulus are simultaneously biased towards the bottom. And, the decrease in the elastic modulus with increasing the value of $e$ gives a higher effect on the free vibration than the decrease in the density along with the value of $e$.

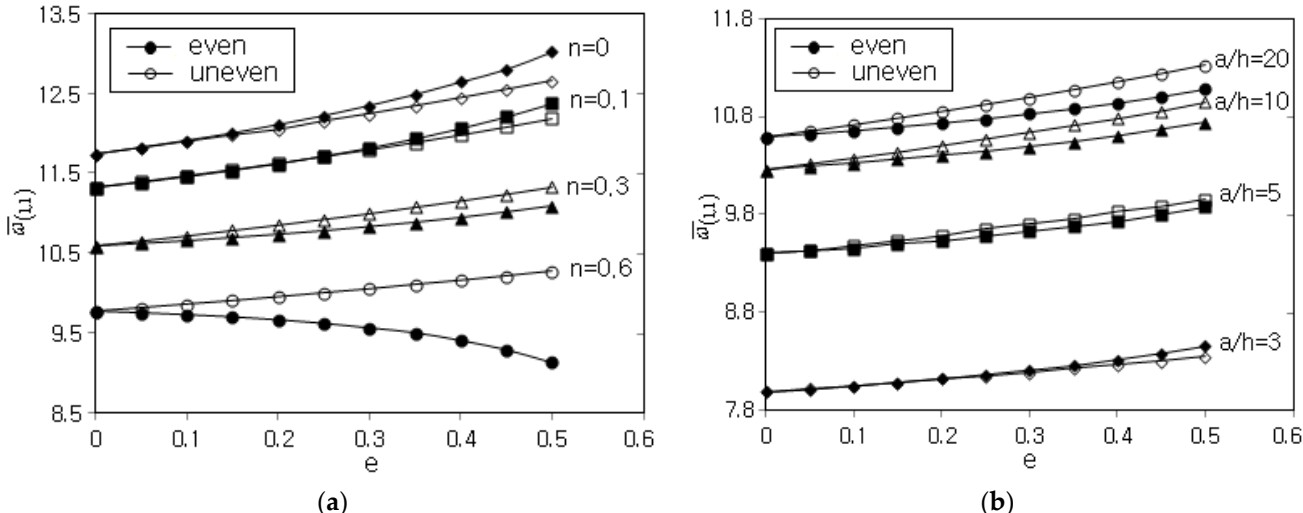

**Figure 7.** Dependence of the calibrated fundamental frequency on the porosity parameter: (**a**) for different ceramic indices ($a/h = 20$), (**b**) for different width-thickness ratios $n = 0.3$.

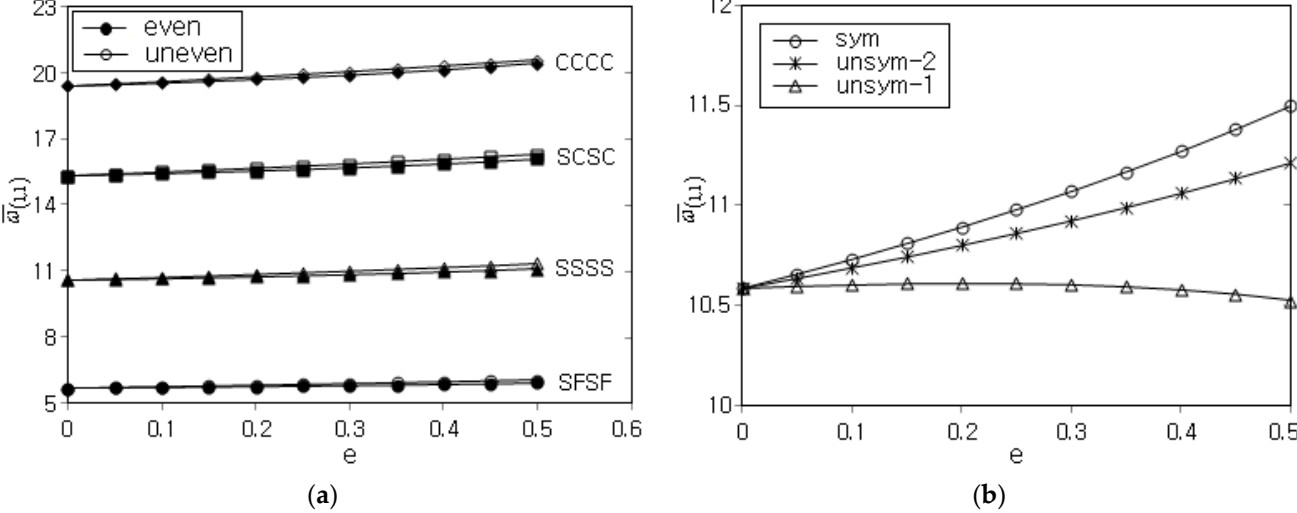

**Figure 8.** Dependence of the calibrated fundamental frequency on the porosity parameter ($a/h = 20, n = 0.3$): (**a**) for different boundary conditions, (**b**) for cosine-type.

## 5. Conclusions

The static bending and free vibration of FG porous plates were numerically analyzed by making use of the (3,3,2) hierarchical model and 2D NEM. The power-law function was adopted to express the thickness-wise metal and ceramic volume fractions, and five different porosity distributions were considered. The proposed numerical method was demonstrated and verified through the benchmark experiment. Moreover, the central deflection and the fundamental frequency were parametrically investigated with respect to the combination of the material composition distribution, the porosity distribution and the $w/t$ ratio, and to the boundary condition. From the numerical results, the following main observations are drawn:

- The maximum relative difference in the calibrated central deflections and fundamental frequencies between the present numerical method and the reference is 5.336%.
- The calibrated central deflection increases with the porosity parameter, and the calibrated fundamental frequency also increases with the porosity parameter, except for the even distribution with $n = 0.6$.

- The even porosity distribution leads to higher central deflection than the uneven porosity distribution, and this relative difference between two porosity distributions uniformly increases with the porosity parameter and the ceramic index, but it is not affected by the plate width–thickness ratio.
- The relative order in the magnitude of calibrated fundamental frequency is dependent on the ceramic index and the width–thickness ratio. The relative difference between even and uneven porosity distributions uniformly increases with the ceramic index and the width–thickness ratio.
- The order in the magnitude of calibrated central deflection among four boundary conditions is SFSF > SSSS > SCSC > CCCC, but this relative order becomes completely reversed for the calibrated fundamental frequency.
- Regarding cosine-type porosity distributions, the unsym-1 provides the highest calibrated central deflection while the sym leads to the lowest level, but this relative order becomes completely reversed for the calibrated fundamental frequency.
- The order in the peak axial stress is unsym-1 > sym > unsym-2 and the order in the peak shear stress is unsym-1 > unsym-2 > sym, respectively.

The current work dealt with metal–ceramic functionally graded plates. Thus, the extension of current work to more advanced materials, such as carbon nanotubes (CNTs) and CNT-reinforced composites, would be worthwhile, and represents a topic for future work.

**Funding:** This work was supported by a National Research Foundation of Korea (NRF) grant funded by the South Korean government (MSIT, 2020R1A2C1100924).

**Conflicts of Interest:** The author declares no conflict of interest.

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
