# Peer review of "Natural Element Static and Free Vibration Analysis of Functionally Graded Porous Composite Plates"

_applsci, doi:10.3390/app122211648_

Round 1

Reviewer 1 Report

The author wrote the manuscript on “Natural element static and free vibration analysis of functionally graded porous composite plates”. The results seems interesting.

However, the authors should clarify before publishing. Some minor observations are:

1.              1. What is n and N in fig. 1 (a)?

2.              2. How fig. 1 (b) was plotted?

3.           3. Table 2 and 3 results shows that the present value is close to the reference values. What is the novelty?

4.              4. In the conclusion, the novelty should be highlighted.

Author Response

Please refer to the attached the response to reviewers' comments (1).

Reviewer 2 Report

·             In the results section, the author needs to analyse the finding by giving reasons for each fact. Please explain every point?

·             The paper should be overviewed against the grammatical error.

·             For more contribution, the authors should compare their results with the related results in other published works such as Longitudinal vibration and instabilities of carbon nanotubes conveying fluid considering size effects of nanoflow and nanostructure Longitudinal vibration and stability analysis of carbon nanotubes conveying viscous fluid Optimal vibration control of multi-layer micro-beams actuated by piezoelectric layer based on modified couple stress and surface stress elasticity theories Study on size-dependent vibration and stability of DWCNTs subjected to moving nanoparticles and embedded on two-parameter foundations

·             The range of considered variables must be put in the Abstract.

·             The introduction section lacks logic and does not highlight the shortcomings of available relevant studies and the novelties of the paper.

·             The model validation section needs further refinement to quantify the deviations between the current predictions and previous results and explain their reasons.

·             Quantify the main outcome in the abstract.

·             The abstract should include the novelty and the reason for the importance of this subject.

·             Define all the abbreviations of the equations in the text.

·             All the figures and tables should be with a detailed caption

·             Define all the abbreviations of the equations in the text.

Author Response

Please refer to the response to reviewers' comments (2).

Reviewer 3 Report

The paper is well-written but needs some minor revisions.1-      In the introduction, the applications of the proposed model should be mentioned in the beginning of the section.2-      There are methods used to solve the proposed model that should be included.3-      The motivation for this work should be added to the end of the introduction section.4-      The parameters in the equations in section 2 should be defined clearly.5-      Several typo errors should be revised.6-      The references should be prepared according to the journal style.

Author Response

Please refer to the response to reviewers' comments (3).
